# Pediatric Carotid Injury after Blunt Trauma and the Necessity of CT and CTA—A Narrative Literature Review

**DOI:** 10.3390/jcm13123359

**Published:** 2024-06-07

**Authors:** Lukas Krüger, Oliver Kamp, Katharina Alfen, Jens Theysohn, Marcel Dudda, Lars Becker

**Affiliations:** 1Department of Trauma Surgery, Hand and Reconstructive Surgery, University Hospital Essen, 45147 Essen, Germany; lukas.krueger@uk-essen.de (L.K.); oliver.kamp@uk-essen.de (O.K.); 2Department of Pediatrics I, Neonatology, Pediatric Intensive Care Medicine and Pediatric Neurology, University Hospital Essen, 45147 Essen, Germany; katharina.alfen@uk-essen.de; 3Institute for Diagnostic and Interventional Radiology and Neuroradiology, University Hospital Essen, 45147 Essen, Germany; jens.theysohn@uk-essen.de; 4Department of Orthopedics and Trauma Surgery, BG-Klinikum Duisburg, 47249 Duisburg, Germany

**Keywords:** blunt carotid injury (BCI), blunt trauma, pediatrics, computed tomography (CT), computed tomography angiography (CTA), radiation, radiation dose, radiation risks, carcinogenesis, ultrasound

## Abstract

**Background:** Blunt carotid injury (BCI) in pediatric trauma is quite rare. Due to the low number of cases, only a few reports and studies have been conducted on this topic. This review will discuss how frequent BCI/blunt cerebrovascular injury (BCVI) on pediatric patients after blunt trauma is, what routine diagnostics looks like, if a computed tomography (CT)/computed tomography angiography (CTA) scan on pediatric patients after blunt trauma is always necessary and if there are any negative health effects. **Methods:** This narrative literature review includes reviews, systematic reviews, case reports and original studies in the English language between 1999 and 2020 that deal with pediatric blunt trauma and the diagnostics of BCI and BCVI. Furthermore, publications on the risk of radiation exposure for children were included in this study. For literature research, Medline (PubMed) and the Cochrane library were used. **Results:** Pediatric BCI/BCVI shows an overall incidence between 0.03 and 0.5% of confirmed BCI/BCVI cases due to pediatric blunt trauma. In total, 1.1–3.5% of pediatric blunt trauma patients underwent CTA to detect BCI/BCVI. Only 0.17–1.2% of all CTA scans show a positive diagnosis for BCI/BCVI. In children, the median volume CT dose index on a non-contrast head CT is 33 milligrays (mGy), whereas a computed tomography angiography needs at least 138 mGy. A cumulative dose of about 50 mGy almost triples the risk of leukemia, and a cumulative dose of about 60 mGy triples the risk of brain cancer. **Conclusions:** Given that a BCI/BCVI could have extensive neurological consequences for children, it is necessary to evaluate routine pediatric diagnostics after blunt trauma. CT and CTA are mostly used in routine BCI/BCVI diagnostics. However, since radiation exposure in children should be as low as reasonably achievable, it should be asked if other diagnostic methods could be used to identify risk groups. Trauma guidelines and clinical scores like the McGovern score are established BCI/BCVI screening options, as well as duplex ultrasound.

## 1. Introduction

BCIs/BCVIs in pediatric trauma have not been extensively researched. To date, there have only been a few publications on this subject. The reason for this might be that BCI/BCVI in pediatric trauma is very rare. A study on the US national pediatric trauma registry revealed an overall incidence of 0.03%. Only 15 out of over 57,000 children had a BCI/BCVI after blunt trauma [1]. A study on the German trauma registry (TraumaRegisterDGU^®^) of the German Society for Trauma Surgery (Deutsche Gesellschaft für Unfallchirurgie, DGU) could only detect an overall prevalence of 0.5%. Only 48 BCI/BCVI cases in 42 out of 8128 severely injured pediatric patients with blunt trauma were registered [2]. 

The “Gold Standard” for screening patients with suspected BCI/BCVI is by CTA followed by alternatives such as magnetic resonance angiography (MRA) or digital subtraction angiography (DSA) [3,4]. In addition to these instrumental diagnostic tools, clinical scores like the McGovern score have been published, which shows a sensitivity of 81% and specificity of 71.3% in BCI/BCVI prediction [4]. Also, ultrasound was named as an excellent possible method to screen for BCI/BCVI [5,6]. Because there are no guidelines for routine diagnostic work-up in pediatric BCI/BCVI, the establishment of a new diagnostic standard is urgently needed.

Many studies show that even minimal radiation exposure in children could lead to more cases of pediatric cancer. Direct evidence from epidemiological studies shows that the organ doses of a common CT study result in an increased risk of cancer. Two or three scans lead to a cumulative dose in the range of 30 to 90 millisieverts (mSv) [7,8]. Therefore, we should critically examine the routine use of radiation to diagnose BCI/BCVI.

This narrative review aimed to investigate how often CT scans are performed in pediatric trauma, how often CTA is used to detect BCI/BCVI after blunt trauma, how many scans with negative results are performed and if it is possible to define risk groups based on clinical scores or ultrasound findings to prevent children from being exposed to radiation.

## 2. Materials and Methods

### 2.1. Definitions

Blunt trauma is defined as any physical impact on the human body that may lead to any injury of the whole body. Any penetrating or sharp traumata were excluded from this research. BCI/BCVI is defined as an injury of the internal carotid artery, common carotid artery and vertebral artery forced by longitudinal stretching, acceleration–deceleration, rotation and hyperextension of the neck, stressing the craniocervical vessels through blunt trauma. Children are defined as any patients under the age of 18. In this study, all other synonyms for children were included.

### 2.2. Search Strategy

Two clinical questions are discussed in this review: Q1: How frequent is BCI/BCVI in pediatric patients after blunt trauma, and what does routine diagnostics look like? Q2: Is a CT/CTA scan of pediatric patients after blunt trauma always necessary and are there any negative health effects?

To find an answer to Q1, this literature review included publications that deal with pediatric blunt trauma and the diagnostics of BCI and BCVI. Publications dealing with non-pediatric BCI/BCVI patients were included to check whether there are any diagnostic or therapy guidelines for adults. For literature research, Medline (PubMed) and the Cochrane library were used. MeSH terms (Medical Subject Heading) “Blunt carotid injury” (BCI) AND “pediatric blunt trauma” were used in the Medline research. The Cochrane library was used to check if there are more potentially relevant publications on this topic that are not included in Medline.

Furthermore, for answering Q2, publications on the risk of radiation exposure for children were included in this study: different diagnostic modalities were correlated with ultrasound and clinical scores (e.g., CTA, MRA, DSA). We aimed to determine whether risk group stratification based on clinical scores and ultrasound can reduce radiation exposure in children. MeSH terms were “pediatric blunt trauma” AND “CT”, OR “CTA”; “radiation dose” AND “radiation risks” AND “carcinogenesis”; and additionally “ultrasound”.

### 2.3. Selection Criteria

Due to a lack of research articles, we included reviews, systematic reviews, case reports and original studies for this systematic review. Literature research included all publications in the English language between 1999 and 2020. Due to an enormous change in imaging diagnostics, publications prior to 1999 were excluded. Publications with titles that were irrelevant were excluded immediately. After that, all abstracts of included publications were read, and irrelevant publications were excluded again. For all relevant publications, the full-text version was considered. If the full-text version was not available, the publications were excluded. For an overview of the review process, see the preferred reporting items for systematic reviews and meta-analyses (PRISMA) in Figure 1 and Figure 2.

## 3. Results

After analyzing all included publications dealing with pediatric BCI/BCVI, we found that most showed an overall incidence between 0.03 and 0.5% of confirmed BCI/BCVI cases due to pediatric blunt trauma [2,3,9,10,11,12,13]. Only two individual studies found a higher incidence of up to 0.9% and 1.1% [14,15]. About 2.7–16% of pediatric blunt trauma patients underwent imaging procedures to detect BCI/BCVI. Of these patients, 64–71% underwent CTA [3,4,16]. Only a few patients underwent other diagnostics like MRA (23.3%) or DSA (11.3%) [3]. Only 0.17–1.2% of all CTA scans resulted in the diagnosis of a BCI/BCVI [3,4,16]. About 76% of all carotid and 67% vertebral arteries were restudied with arteriography 7–10 days after the injury [17].

### 3.1. Screening Tools

Publications before 2018 mainly used modified Memphis criteria [18] (screening criteria based on adult patients) to screen pediatric patients and decide whether CTA was necessary or not. Modified Memphis criteria classify basilar skull fracture with involvement of petrous bone, basilar skull fracture with involvement of the carotid canal, Le Fort II or III fracture pattern, cervical spine fracture, Horner’s syndrome, neck soft-tissue injury (like seatbelt sign, hanging or hematoma) and focal neurological deficit not explained by imaging as screening criteria for BCI/BCVI (Table 1). If one of these screening criteria is met, the recommendation to perform further work-up with angiographic imaging is given [18]. However, these studies showed that carotid or vertebral imaging was performed only in 16.5% of cases with at least one risk factor. Nevertheless, imaging was performed in 1.69% of cases even though no risk factors were detected, and 3 out of 44 scans (6.8%) detected a BCI/BCVI [3]. Other clinical scores, namely, the Denver [19], EAST [20] and Utah scores [21], use other screening tools to detect BCI/BCVI. Screening tools are also summarized in Table 1. A study from the University of Missouri, Columbia, has shown that modified Memphis criteria misclassified 28.6% of all pediatric trauma cases. Also, the Denver, EAST and Utah scores misclassified 28.6%, 33.3% and 47.6%, respectively. Based on the Utah score, the newly created McGovern score was presented with a sensitivity of 81% and a specificity of 71.3% to detect BCI/BCVI correctly. The McGovern score comprises six elements that were identified as risk factors for BCI/BCVI: Glasgow Coma Scale < 8, focal neurological deficit, carotid canal fracture, petrous temporal bone fracture, cerebral infarction on CT, and a motor vehicle accident as a mechanism of injury (MOI) [4]. Other risk factors like a seatbelt sign were not associated with BCI/BCVI [22]. Until now, the McGovern score has not been validated by a second study. Another study of 2019 even showed that the recently added MOI of a motor vehicle accident had no significant correlation with BCI/BCVI [23]. Important results are summarized in Table 2.

CT is often used for diagnostics after blunt trauma to identify injuries because of its fast, extensive and precise results. In total, 52.5% of all pediatric polytrauma patients receive a CT for primary diagnostic work-up [26]. A study on the TraumaRegisterDGU^®^ has shown that in the control and BCI/BCVI groups, children underwent immediate head/neck CT in 85.3% vs. 94.4%, or whole-body CT in 64.6% vs. 86.1% [2]. As diagnosis by imaging is included in almost all scores, a CT of the head and neck is necessary for every patient to be screened. Scores that do not require CT or DSA to screen for BCI/BCVI have not been published to date.

### 3.2. ATLS Guidelines for CT Scans on Pediatric Trauma

The tenth edition of Advanced Trauma Life Support (ATLS) provides new guidelines (2018) on when CT scans on pediatric trauma should be performed. Following this algorithm, 58.3% of the population should not receive a CT at all. A group of 27.7%, including patients with a history of loss of consciousness (LOC) or history of vomiting, severe mechanism of injury, or severe headache, should be observed first, and the decision to perform a CT should be taken based on other clinical factors. The indication could be based on physician experience, multiple vs. isolated findings, worsening symptoms or worsening signs after emergency department observation, and parental preference. Only patients with GCS = 14, other signs of altered mental status or basilar skull fracture signs should receive a CT immediately. This group includes only 14% of all pediatric traumata [27] (Figure 3).

### 3.3. Scandinavian Guidelines for Initial Management of Pediatric Head Trauma

Astrand et al. published Scandinavian guidelines for initial management of minor and moderate head trauma of children in 2016. Following their flow chart, pediatric patients with moderate head trauma, GCS 9–13, should receive a CT scan immediately. Patients with mild head trauma are divided into risk groups. The high-risk group is defined as a GCS of 14–15 and focal neurological deficit, or post-traumatic seizures, or clinical signs of skull base fracture or depressed skull fracture. The high-risk group should also receive a CT scan immediately. Medium risk is defined as a GCS of 14 or 15 with an LOC > 1 min, or anticoagulation or coagulation disorder. The medium-risk group should be clinically observed for over 12 h. A CT scan is only considered as an alternative in this group. Low risk is defined as a GCS of 15 with post-traumatic amnesia, or severe/progressive headache, or abnormal behavior according to the child’s guardian/s, or vomiting ≥ 2 times, or suspected/brief LOC, or preexisting cerebral shunt, or if age < 2 years, a large temporal or parietal scalp hematoma or irritability. A clinical observation for 6 h is considered in the low-risk group, and a CT scan is only advised for patients with multiple risk factors [24] (Figure 4).

### 3.4. Imaging Procedures and Their Radiation Risks

As restricted use of diagnostic radiation in children should be standard, we need to examine the dose of radiation exposure of CT scans, the radiation risk and the alternatives to CT. In children, the median volume CT dose index on a non-contrast head CT is 33 milligrays (mGy) [28]. A CT of the skull or facial bones needs 27–37 mGy, a scan of the neck 19–26 mGy and a scan of petrous bones 42–67 mGy depending on the age group [29]. A computed tomography angiography needs at least 138 mGy [30]. Radiation exposure is associated with a higher cancer incidence. In 2007, the Center for Radiological Research, Columbia University Medical Center, New York, found direct evidence from epidemiological studies that the organ doses of a standard CT correspond to an increased risk of cancer [7]. A study from the Netherlands reported that the cumulative brain dose of a pediatric brain CT scan was 38.5 mGy and was statistically significantly associated with an increased brain tumor risk [31]. The Institute of Health and Society and the Northern Institute for Cancer Research, Newcastle, noted the correlation between radiation dose from CT scans and leukemia and brain tumors. The use of CT scans in children with cumulative doses of about 50 mGy almost triples the risk of leukemia, and doses of about 60 mGy triples the risk of brain cancer [32]. Furthermore, age at the time of exposure and the lifetime attributable risk for children could be identified as an essential risk factor that makes them more vulnerable to radiation exposure than adults [33]. In cases where children received CTs at hospitals without a pediatric trauma center, the median effective radiation dose was two times higher [34]. The key facts are summarized in Table 3.

### 3.5. Ultrasound as an Alternative Diagnostic Tool

In a case report on a 12-year-old boy, duplex ultrasound was mentioned as a possible imaging tool to detect BCI/BCVI. Before treatment, a CTA was conducted to confirm the diagnosis made by ultrasound and detect the injuries’ extent [5]. The same publication illustrates a diagnostic algorithm by ATLS from 2005 for asymptomatic pediatric BCI/BCVI patients. There, duplex ultrasound is shown as the first diagnostic step for asymptomatic patients. This algorithm can no longer be found in the newest edition (2018) of ATLS. It could be shown that transcranial doppler measurement was significantly associated with the adult blunt cervical vascular injury status. It was suggested that transcranial doppler sonography could be a viable bedside screening tool for trauma [35]. A study on the meaningful use of ultrasound on pediatric BCI/BCVI has not been published yet.

## 4. Discussion

Taking these results into consideration, it is confirmed that pediatric BCI/BCVI is a rare injury. The number of publications on adult BCI/BCVI after blunt trauma is low and on pediatric BCI/BCVI even lower. Nevertheless, it can be seen that BCI/BCVI diagnostic work-up mostly follows the same routine. After patients reach the hospital, an extensive physical examination should be conducted. Asymptomatic patients could be screened by duplex ultrasound to detect BCI/BCVI and prevent them from unnecessary radiation exposure. Due to better anatomic and physical conditions in pediatric patients, ultrasound should be easier to perform than in adult patients. A positive BCI/BCVI suspicion on duplex ultrasound needs clarification via CTA. According to the ATLS guidelines, only patients with GCS = 14, other signs of altered mental status or basilar skull fracture signs should receive a CT scan immediately. All other patients should be observed, and indications for a CT scan should be set narrowly [27]. If patients show neurological symptoms, a CT scan is irreplaceable.

Assuming that a patient could have a BCI/BCVI, suspected by clinical examination, ultrasound or CT scan, the indication for a CTA scan should be checked carefully. Currently, the McGovern score is the only score with a comparatively high sensitivity and specificity for BCI/BCVI detection. As one study has raised concerns about the correlation of the MOI of a motor vehicle accident and BCI/BCVI incidence, additional studies’ independent validation of the McGovern score is necessary prior to implementation in routine diagnostics.

Considering the data regarding radiation exposure and the risk of cancer, many publications have proven the correlation between radiation exposure and the probability of the appearance of cancer. Because a cumulative radiation dose of about 50 mGy is sufficient to triple the risk of leukemia and a dose of about 60 mGy to triple the risk of brain cancer, one CTA or two CT scans are enough to reach or even exceed this cumulative radiation threshold dose in children [32]. Taking a critical look at the analyzed studies, we found that 2.7–16% of all pediatric blunt trauma patients underwent imaging procedures to detect BCI/BCVI. Due to the low number of performed scans, the indication for scans was mainly decided after careful consideration. However, in the case where an imaging procedure was used for diagnosis, it was a CTA in up to 71% of cases. Whether these indications were derived by scores or just by expert opinion is unclear. Nevertheless, only 0.17–1.2% of these CTA scans allowed for a positive BCI/BCVI diagnosis.

Following the ATLS guidelines, only 14% of all trauma patients should receive a CT scan immediately, while 27.7% of all trauma patients might receive a CT scan at a later point after careful observation. In reality, 52.5% of pediatric patients receive a CT scan after trauma significantly exceeding the recommendation by ATLS. The stringent use of CT guidelines in the management of pediatric trauma could reduce this number to 42.8% [26]. 

Validation of the Scandinavian guidelines was recently achieved in 2020. The results show that the guidelines have a very high negative predictive value of 99.9% to identify children without traumatic brain injury (TBI), and a sensitivity of 92.3% for the detection of TBI. It was proven that the use of the Scandinavian guidelines would potentially reduce the use of CT. In this study, 25.0% of the CTs were performed in children with minimal head injury. It was also shown that the guidelines have a high negative predictive value of 96.9% for traumatic findings on CT. The study validation also showed that unnecessary CT scans could be avoided in the mild low-risk head injury group, which accounts for 51.1% of the CT scans. Furthermore, following the guidelines could have saved 76.1% of all CTs performed on these patients [25]. 

The slim database published so far makes it challenging to reach a generally applicable statement. Almost all published studies, except the Scandinavian guidelines, were conducted in the US, making it difficult to apply the results to other countries and healthcare systems. Other guidelines, expert opinions and other legal statuses may affect the results analyzed in other countries.

Our clinical observations have shown that imaging procedures are very frequently used in pediatric trauma care. There is often a concern about not being able to diagnose a BCI/BCVI at an early stage and thus delaying a possible intervention. The knowledge of the possible serious long-term neurological consequences for the children often leads the responsible physician to have a CTA performed to be on the safe side. The possible long-term consequences of radiation exposure are disregarded, as they have no consequences for the treating physician at this stage.

### Limitations

The review includes publications from the period 1999–2020, during which there were significant changes in the medical field, including a change in the number of cases, the possibilities of different diagnostics and a reduction in the radiation exposure of modern equipment. This heterogeneity of the data makes it sometimes difficult to make direct comparisons. In addition, the different publications show different qualities in the scientific research work. This review draws on retrospective data. As there are only a few scientific publications on this topic and the number of cases is also very low, all publications relevant to the topic were included wherever possible. It was not screened for publication bias. Nevertheless, this review provides a valuable and comprehensive up-to-date overview that could be helpful for further research work in this field.

## 5. Conclusions

We have shown that BCI/BCVI in pediatric trauma is a rare injury. The “Gold Standard” diagnostic tool for trauma patients is the plain CT scan, and for BCI/BCVI, it should be the CTA. Each individual radiological examination is accompanied by the acceptance of certain radiation risks. A CTA scan of the brain may triple the lifetime risk of brain cancer. As there are many negative and unnecessary scans, radiation-containing imaging should be used with caution and should be significantly minimized. Duplex ultrasound and the clinical McGovern score could help to identify risk groups for BCI/BCVI and could contribute to a decreasing number of unnecessary CTA scans. The McGovern score needs to be validated by other studies. CT guidelines for the management of pediatric trauma and BCI/BCVI could reduce the number of radiation-containing scans, especially in hospitals without a pediatric trauma center.

## Figures and Tables

**Figure 1 jcm-13-03359-f001:**
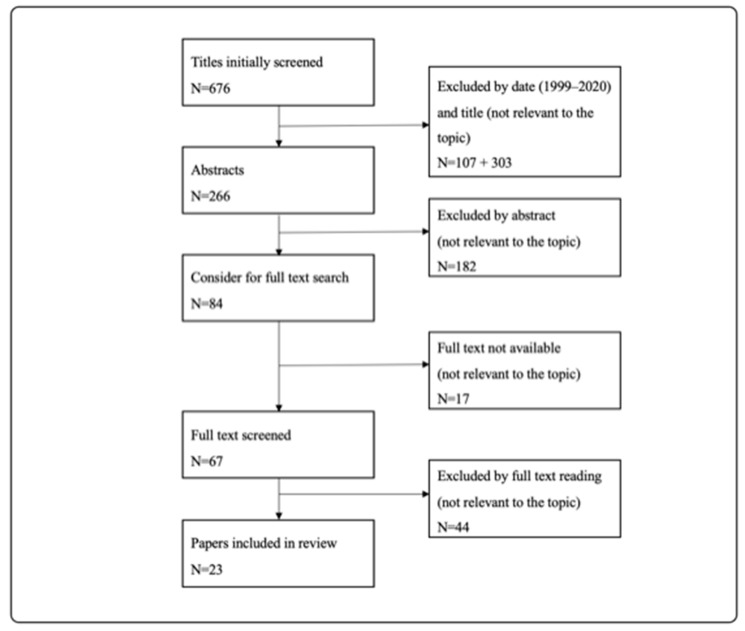
PRISMA (preferred reporting items for systematic reviews and meta-analyses) diagram explaining the review process for research question 1: How often is BCI/BCVI on pediatric patients after blunt trauma and what does routine diagnostic looks like?

**Figure 2 jcm-13-03359-f002:**
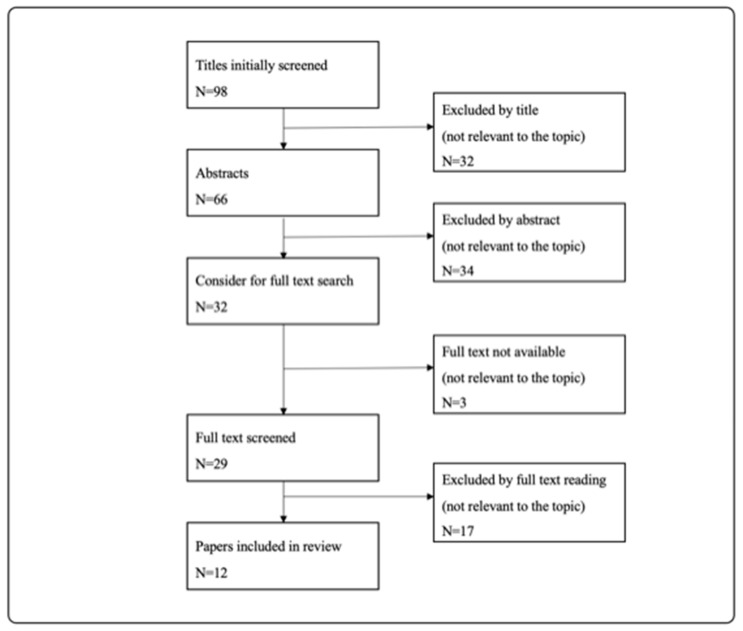
PRISMA (preferred reporting items for systematic reviews and meta-analyses) diagram explaining the review process for research question 2: Is a CT/CTA scan on pediatric patients after blunt trauma always necessary and are there any negative health effects?

**Figure 3 jcm-13-03359-f003:**
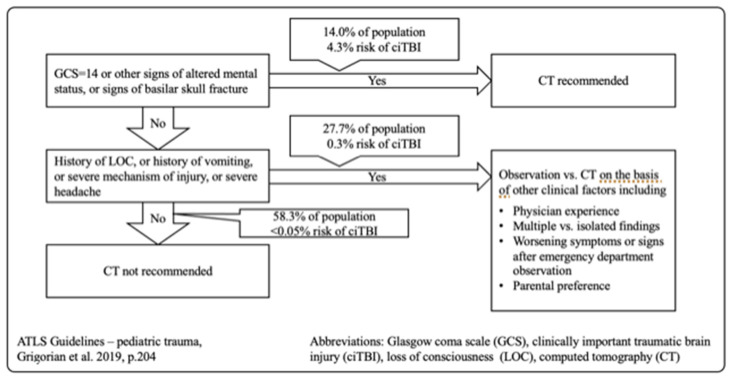
Pediatric Emergency Care Applied Research Network (PECARN) Criteria for Head CT [23].

**Figure 4 jcm-13-03359-f004:**
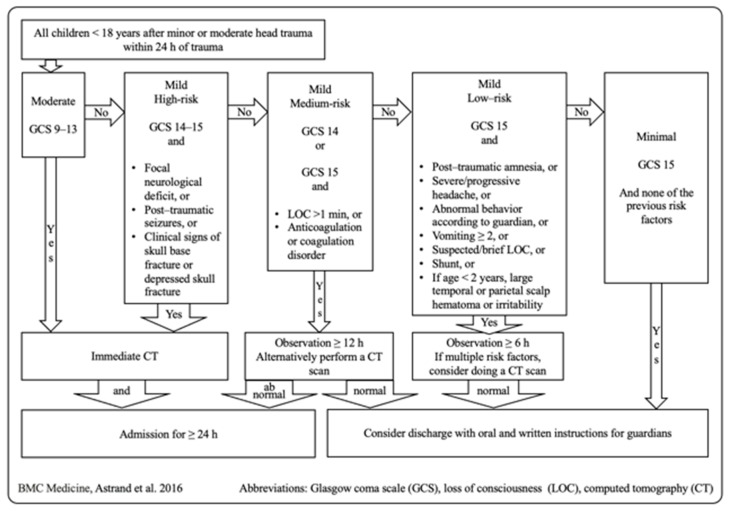
Scandinavian guidelines for initial management of minor and moderate head trauma in children [26].

**Table 1 jcm-13-03359-t001:** Summary of all screening criteria for BCI/BCVI.

Denver Criteria	EAST Criteria	Modified Memphis Criteria	Utah Criteria	McGovern Criteria
Focal neurological deficit	Cervical hyperextension associated w/displaced midface or complex mandibular fracture or closed head injury consistent with diffuse axonal injury	Basilar skull fracture with involvement of petrous bone	GCS score ≤ 8 (1Pt.)	GCS score ≤ 8 (1Pt.)
Arterial hemorrhage	Anoxic brain injury due to hypoxia as a result of squeezed arteries	Basilar skull fracture with involvement of the carotid canal	Focal neurological deficit (2Pt.)	Focal neurological deficit (2Pt.)
Cervical bruit in patients < 50 yrs	Seatbelt abrasion or other soft-tissue injury resulting in swelling or altered mental status	Le Fort II or III fracture pattern	Carotid canal fracture (2Pt.)	Carotid canal fracture (2Pt.)
Expanding neck hematoma	Cervical vertebral body fracture or carotid canal fracture in proximity to the internal carotid or vertebral arteries	Cervical spine fracture	Petrous temporal bone fracture (3Pt.)	Petrous temporal bone fracture (3Pt.)
Neurological exam findings inconsistent w/head CT scan		Horner’s syndrome	Cerebral infarction on CT (3Pt.)	Cerebral infarction on CT (3Pt.)
Cerebrovascular accident on follow-up head CT scan not seen on initial head CT scan		Neck soft-tissue injury (seatbelt sign, hanging or hematoma)		MOI (2Pt.)
Presence of Le Fort II or III fractures		Focal neurological deficit not explained by imaging		
Cervical spine fracture w/subluxation				
C1–3 cervical spine fracture				
Cervical spine fracture extending into the transverse foramen				
Basilar skull fracture w/carotid involvement				
Diffuse axonal injury w/GCS score < 6				
Hypoxic ischemia due to squeezed arteries				

For Denver, EAST (The Eastern Association for the Surgery of Trauma)and modified Memphis criteria, further work-up with angiographic imaging is recommended if any of the listed criteria are met. For Utah and McGovern criteria, a score ≥ 3 points on both scales signifies high risk of BCI/BCVI and indicates angiography.

**Table 2 jcm-13-03359-t002:** Summary of results based on BCI/BCVI incidence and diagnostic criteria (clinical question 1).

Publication	Number of Cases	BCI/BCVI Incidence	Diagnostic Tool	Classified BCI/BCVI Correct	Misclassified BCI/BCVI
Astrand R. 2016 et al. [24]	118,265	0.18–0.3% (212–355)	GCS CT	NA	NA
Azarakhsh, N. 2013 et al. [3]	5829	0.4% (23)	Memphis criteria	20 (87%)	3 (13%)
Ciapetti, M. 2010 et al. [18]	266	2% (6)	Modified Memphis criteria	6 (100%)	0
Cuff, R. 2005 et al. [5]	1	NA	Duplex ultrasound	1 (100%)	0
Grigorian, A. 2019 et al. [23]	69,149	0.2% (109)	NA	NA	NA
Herbert, J.P. 2018 et al. [4]	12,614	0.17% (21)	Denver, modified Memphis, Eastern Association for the Surgery of Trauma (EAST), Utah, McGovern—screening score	15 (71%) 15 (71%) 13 (67%) 11 (52%) 17 (81%)	6 (29%) 6 (29%) 7 (33%) 10 (48%) 4 (19%)
Jones, T.S. 2012 et al. [10]	14,991	0.3% (45)	NA	NA	NA
Kerwin, A.J. 2001 et al. [15]	2331	1.1% (25)	NA	NA	NA
Kraus, R.R. 1999 et al. [11]	5835	0.27% (16)	NA	NA	NA
Leraas, H.J. 2019 et al. [22]	422,181	0.19% (809)	Denver, Memphis—screening score	NA	NA
Lew, S.M. 1999 et al. [1]	57,000	0.03% (15)	NA	NA	NA
Li, W. 2010 et al. [9]	1,633,126	0.05% (842)	NA	NA	NA
Ravindra, V.M. 2017 et al. [21]	411	5.4% (22)	Utah screening score	18 (83.4%)	4 (16.6%)
Singh, R.R. 2004 et al. [6]	NA	NA	Angiography CT/CTA MRI/MRA Ultrasound	Up to 100% NA 95–99% NA	NA
Sönnerqvist, C. 2021 et al. [25]	43,025	NA	Scandinavian guidelines for initial management of minor and moderate head trauma in children (SNC-G)	negative predictive value for ciTBI (99.9%), sensitivity for detection of ciTBI (92.3%), negative predictive value for traumatic findings on CT (96.9%)	NA

Abbreviations: blunt carotid injury (BCI), blunt cerebrovascular injury (BCVI), Glasgow coma scale (GCS), computed tomography (CT), computed tomography angiography (CTA), magnetic resonance imaging (MRI), magnetic resonance angiography (MRA), clinically important traumatic brain injury (ciTBI).

**Table 3 jcm-13-03359-t003:** Summary of key facts for imaging procedures and their radiation risks (clinical question 2).

Publication	Results
Brenner, D. J. 2007 et al. [7]	The organ doses of a common CT scan result in an increased risk of cancer. Two or three scans, resulting in a dose in the range of 30 to 90 mSv
Goodman, T.R. 2019 et al. [33]	Age at the time of exposure and the lifetime attributable risk for children are essential risk factors that make them more vulnerable to radiation exposure than adults
Journy, N.M.Y. 2018 et al. [29]	A CT scan of the skull or facial bones needs 27–37 mGy, a scan of the neck 19–26 mGy and a scan of petrous bones 42–67 mGy
McGrew, P.R. 2018 et al. [26]	52.5% of all pediatric polytrauma patients receive a CT for primary diagnostic work-up
Meulepas, J.M. 2019 et al. [31]	The cumulative brain dose of a pediatric brain CT scan was 38.5 mGy and was statistically significantly associated with brain tumor risk
Nabaweesi, R. 2018 et al. [34]	Children received CTs at hospitals without a pediatric trauma center, median effective radiation dose was two times higher
Pearce, M.S. 2012 et al. [32]	Radiation dose of 50 mGy is sufficient to triple the risk of leukemia A dose of about 60 mGy is sufficient to triple the risk of brain cancer One CTA or two CT scans are enough to reach or even exceed the cumulative radiation threshold dose in children
Sadigh, G. 2018 et al. [28]	In children, the median volume CT dose index on a non-contrast head CT is 33 mGy
Schneider, T. 2017 et al. [30]	A computed tomography angiography needs at least 138 mGy

Abbreviations: computed tomography (CT), computed tomography angiography (CTA), milligrays (mGy).

## Data Availability

No new data were created or analyzed in this study. Data sharing is not applicable to this article.

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
