# Peer review of "Pediatric Carotid Injury after Blunt Trauma and the Necessity of CT and CTA—A Narrative Literature Review"

_jcm, 2024, doi:10.3390/jcm13123359_

Round 1
Reviewer 1 Report
Comments and Suggestions for Authors
Dear Authors,
Very rare but interesting topic. Figures are also well-described as useful guidelines.
Best regards,
Reviewer
Author Response
Thank you very much for your kind review and feedback.
Reviewer 2 Report
Comments and Suggestions for Authors
This is a review article over the topic blunt carotid traumatic injury in children. The authors are trying to find using a search over the last 25 years of literature on this topic. Their goal is to find out the relative incidence of this type of injuries and what are the preferred diagnostic tools for this type of injury in pediatric patients. In my opinion they only succeeded to find an answer for the first of the two questions, the incidence of the BCI in literature. For the second question, their results are rather confusing I cannot find an direct link between the presented results and the conclusion of the study.
The introduction section offers sufficient information for the reader to become familiarized with the subject.
The material and methods section is presenting the strategy of the study in an comprehensive and well planned manner.
Unfortunately, the results section is very difficult to follow. It is rather an interpretation of the results. I recommend re-writing this section and presenting the results only and in a strictly factual manner. Perhaps a table with the results of each study that was considered in this review.
For instance, statements like Line 103 - “Only a few patients underwent other diagnostics like MRA or DSA” is not a scientific result.
In the discussion section the authors offer an interpretation of their findings. However, is very difficult to differentiate from the results section.
The conclusion section offers a recommendation over what should be the approach in pediatric BCI and how often should we expect to see this kind of lesions in children. This recommendations are welcome and might be of use for the clinician, especially when one is trying to reduce the unnecessary irradiation burden.
Line 14 - BCIV – abbreviations should be defined when first mentioned in text
Line 15 - CT/ CTA – abbreviations should be defined when first mentioned in text
Line 41 - DGU - abbreviations should be defined when first mentioned in text
Line 106 - modified Memphis criteria – appropriate citation is required
References format is not as recommended
Author Response
Thank you for your detailed and critical review.We have rewrote the "results" and "discussion" sections and added additional tables as recommended.
We have attempted to clarify these parts of the manuscript and hope we have succeeded. In addition, all abbreviations were defined when they first appeared in the text.
Reviewer 3 Report
Comments and Suggestions for Authors
Congratulations for your work on this interesting topic, which presents extremely low incidence on the other hand. Some comments for improvemen:
- By using the title of a systematic review, I would expect a classic systematic review, which is usually accompanied by a meta-analysis. I would suggest of changing the title to "narrative" review, as you presente your data in descriptive way containing more qualitative data than quantitative data. This could be very obvious if you used the PRISMA framework for systematic reviews, apart the PRISMA flowchart you used in your manuscript. In that case you would not be able to present all relevant data and parts PRISMA statement demands. This is not necessarily bad and does not minimize the value of your work, but it proves that your review is more likely to be described as a narrative one.
- Please provide a paragraph in the discussion presenting the limitations of your work and possible future directions. For example one limitation could be the fact that the studies included are not of high level of evidence.
- Please provide a paragraph in the discussion presenting your opinion and any comments on your findings and comparing them to the current literature.
- You present some quantitative results. Please provide information about any statistics used both in the Methods and the Results part.
- In the search strategy part I feel that you present the outcomes your review investigates. Please separate this part under a subtitle like primary/secondary outcomes
- Similarly, your Results is a long part without any subtitles, as somebody would expect for the results part of a review. Provide description of the flowchart, studies included/excluded and the reasons why they were excluded, main findings, etc. On the other hand, the Discussion part is by far smaller than the results. In this case, I suppose that many data you mention in the results belong to the discussion, where a more critical view of the current literature by you would be more interesting instead of just presenting what current guidelines suggest or mentioning the known fact that studies regarding carotid injury after blunt trauma are few not providing adequate data to reach a unanimous statement.
Author Response
Thank you very much for the intensive and critical review.Regarding the title, we share your critical comment and have adjusted the title to "narrative review" accordingly. We rewrote the section "results" and "discussion", subtitles a paragraph concerning limitations of the study was added.
We really appreciate your review and hope that we have responded satisfactorily to your comments.
Round 2
Reviewer 3 Report
Comments and Suggestions for Authors
Congratulations again for your manuscript. You have made all changes reviewers asked. However, I would like the limitations to be included inthe discussion part. Otherwise, I find the manuscript quite interesting.
Author Response
Thank you again for reviewing the manuscript. We greatly appreciate your critical comments and are pleased that our efforts to improve the manuscript were successful. We have included the section on limitations in the discussion.